# Clients’ Facial Expressions of Self-Compassion, Self-Criticism, and Self-Protection in Emotion-Focused Therapy Videos

**DOI:** 10.3390/ijerph20021129

**Published:** 2023-01-09

**Authors:** Ghazaleh Bailey, Júlia Halamová, Viktória Vráblová

**Affiliations:** Institute of Applied Psychology, Faculty of Social and Economic Sciences, Comenius University in Bratislava, Mlynské luhy 4, 821 05 Bratislava, Slovakia

**Keywords:** emotion-focused therapy, automated facial analysis, iMotions

## Abstract

Clients’ facial expressions allow psychotherapists to gather more information about clients’ emotional processing. This study aims to examine and investigate the facial Action Units (AUs) of self-compassion, self-criticism, and self-protection within real Emotion-Focused Therapy (EFT) sessions. For this purpose, we used the facial analysis software iMotions. Twelve video sessions were selected for the analysis based on specific criteria. For self-compassion, the following AUs were significant: AUs 4 (brow furrow), 15 (lip corner depressor), and the AU12_smile (lip corner puller). For self-criticism, iMotions identified the AUs 2 (outer brow raise), AU1 (inner brow raise), AU7 (lid tighten), AU12_smirk (unilateral lip corner puller), and AU43 (eye closure). Self-protection was combined using the occurrence of AUs 1 and 4 and AU12_smirk. Moreover, the findings support the significance of discerning self-compassion and self-protection as two different concepts.

## 1. Introduction

Over the last decade, the role of self-compassion and self-protection (also known as assertive anger) as antidotes to self-criticism has been increasingly acknowledged as fundamental for mental health [1,2,3,4,5]. Both self-compassion and self-protection play a key role in the emotional change processes [6,7,8] in emotion-focused therapy (EFT). Within EFT theory and research, it is recognized that self-criticism as a form of maladaptive secondary anger turned inside [9,10] can be transformed through the development of primary adaptive emotions such as self-compassion and self-protection [6,7]. The concept of changing negative, maladaptive emotion with positive primary adaptive emotion is a core element of EFT change [6,11]. The EFT sequential model of emotional processing aims to help clients to transform maladaptive emotions with primary adaptive emotions such as self-compassion and self-protection (also known as protective anger) [6]. EFT therapists facilitate self-compassion and self-protection capacities as primary adaptive emotions by implementing specific two-chair interventions [7,12,13,14]. The EFT model distinguishes between two sorts of two-chair dialogues: empty-chair dialogues with a significant other, also known as unfinished business [15,16], and two-chair dialogues used for self-critical dialogues between the critical and the experiential part of the self [14,17]. The marker for an empty-chair dialogue is interpersonal problems such as lingering negative feelings towards a significant other, e.g., a mother or father [11,16]. The marker for a two-chair dialogue is intrapersonal problems such as conflicts within the client’s self in the form of self-criticism [14,18], or as maladaptive anger towards oneself [10]. The purpose of both chair dialogues is to help clients to evoke self-compassion as the ability to internally comfort and soothe themselves and self-protection as a feeling of entitlement to identify and express unmet needs [7,8]. In order for the EFT therapist to guide the client through these processes in a way that is useful and efficient, it is important to understand how self-criticism, self-compassion, and self-protection are expressed within the session [19]. Nonverbal expression of emotional states plays a significant role in EFT [20,21,22]. According to Auszra’s productivity model [20], a client’s manner of emotional processing is marked through seven verbal and nonverbal criteria: attendance, symbolization, congruence, acceptance, regulation, agency, and differentiation. As emotions are primarily demonstrated in the face, a significant way clients express and communicate emotions is over the face [23,24,25]. In accordance with Matsumoto and Hwang [26], understanding the facial expressions of emotions promotes the development of rapport and trust, which is crucial for a therapeutic relationship [27]. Therefore, the therapist’s ability to discern a client’s manner of processing and emotional state through the face is fundamental. The therapist can recognize emotional meaning in the face even when the client cannot express the feeling verbally [23]. The concept that emotions are linked to particular facial expressions goes back to Darwin [28]. In the meantime, there has been an ample amount of research on the facial expressions of basic emotions (anger, fear, happiness, sadness, disgust, contempt, and surprise) in laboratory settings [23,29,30,31]. Furthermore, there is a growing body of research investigating how the face conveys emotions [23,29,31]. The comprehensive facial action coding system (FACS) [32] is the most widely used expression coding system in behavioral sciences. Facial expressions can be analysed in three different ways: Manual coding of facial activity by trained coders [33],Facial Electromyography (fEMG) [34], andAutomatic facial expression analysis by using machine learning algorithms [35].

Developed by Ekman and Friesen [32], FACS is a fully standardized classification system which provides an objective description of anatomical facial action by trained human experts [25,33]. Human coders examine face videos and describe any occurrence of facial expressions as combinations of elementary components called Action Units (AUs). Ekman and Friesen [32] defined a detailed manual of 46 AUs describing different movements of facial muscles including head and eye movements. Each AU number is linked to a FACS that describes a specific facial muscle movement. For example, AU15 is determined by the FACS lip corner suppressor [32]. Although FACS does not describe the meaning of the expression, certain combinations of AUs relate to a displayed emotion [23,33]. For instance, the facial expression of sadness is characterized as the following: inner eyebrows raised (AU1) and drawn together (AU4) and lip corners pulled down (AU15) [23,36]. The AUs for happiness are defined as cheek raiser (AU6) and lip corner puller (AU12) [36]. Furthermore, research is consistently improving in this field, and it has broadened to other emotions in addition to the well-known basic emotions [37,38,39]. Clients’ facial expressions in real psychotherapy sessions are an almost undiscovered area. Thus, this study focuses on the automatic analysis of clients’ facial expressions of self-criticism, self-compassion, and self-protection in EFT video sessions. 

## 2. Facial Expression of Self-Compassion

Whereas research efforts over the last decade have focused on exploring the facial expression of compassion for others [37,38,40,41,42,43], the facial expression of self-compassion remains open. There is a lack of research examining spontaneous compassionate expressions in social interaction. So far, the majority of the studies investigating the facial components of compassion used pictures of actors being asked to pose with compassionate faces [37,40,41,43]. To the best of our knowledge, there are two recent studies investigating the spontaneous facial expression of compassion. Baránková et al. [42] analyzed the facial expressions of participants watching a compassionate moment in a video by manually coding the corresponding FACS and examining which basic emotions fell in line with Ekman and Friesen [23], and which of these were the most representative while watching the video. The results identified participants’ most recurrent facial expressions as the following: lid tightener (AU7), lip corner puller (AU12), eyes closed (AU43), and head tilt right (AU56). Kanovsky et al. [38] amplified the study by analyzing, both manually and automatically, the most common facial expressions of participants watching a variety of compassion-induced videos. Their results show significant agreement with Baránková et al. [42] on the upper face but a different outcome regarding the lower face. In contradiction with the first study, participants watching the most compassionate moment responded with a lip presser instead of pulled lip corners and a left head tilt instead of right one [38]. However, considering compassion as an emotion [37,44], there is a general agreement amongst researchers that compassion is a blended emotion. It is often a mix of sadness [42,43,45,46] and happiness [47]. Therefore, it is significant to distinguish between the facial expressions of sadness and compassion, and compassion and happiness. Furthermore, studies agree that the expression of compassion involves a head inclination [38,42] as well as a genuine smile [41,42]. According to Strauss’ definition of compassion [48], self-compassion is compassion towards oneself. In the light of these considerations, we assume that the in-session facial expression of a client expressing self-compassion is one similar to expressing compassion for others.

## 3. Facial Expression of Self-Criticism

Despite the important role of self-criticism in psychopathology, the majority of research so far has focused on the recognition of facial expressions of emotion by self-critical people according to their level of self-criticism [41,49,50]. For instance, McEwan et al. [41] demonstrate that self-critical people perceive compassionate and smiling faces as distressing rather than enjoyable and supportive. A recent study by Koróniová et al. [51] examined the EMG facial muscle activity of self-critical people viewing imagery of self-compassion, self-protection, and self-criticism depending on the levels of self-criticism. Their results show that the EMG values were significantly higher whilst viewing self-critical imagery and a difference between high and low critics whilst viewing self-compassion imagery. The most important contribution is a new study by [52] exploring the Action Units (AUs) of participants expressing self-criticism in a two-chair dialogue using the computer software iMotions. IMotions is a software that detects Facial Action Units automatically [53]. Their findings propose the facial expression of self-criticism including the action units AU 2, 5, 14, 24, 26, and 43. The authors emphasize that the results are associated with the expressions of contempt, fear, embarrassment, and shame. In line with this, within the literature, self-criticism is defined by intense self-judgement, self-devaluation, self-condemnation [54,55], and self-contempt [56]. To date, there is ample research identifying the facial expression of contempt comprising AU12 (unilateral lip corner raise, also known as a smirk) and AU14 (dimpler) [57,58]. In addition, in EFT theory, self-criticism is perceived as problematic anger, a secondary emotion that covers more vulnerable emotions such as shame [9,10,56]. Based on Ekman and Friesen (2003), the angry face is characterized by lowered and drawn together eyebrows, tensed eyelids, staring eyes, and lips tightly pressed together or parted in a square shape. The two types of an angry mouth occur through different intentions. While the lips are pressed during physical violence and controlled anger, the open mouth occurs during shouting or verbalizing the anger. Anger can also be shown only in particular parts of the face such as the mouth or the brows [23]. Regarding facial AUs, anger is defined by AU 4 and 5 and/or 7, 22, 23, and 24 [36]. Considering that self-criticism is linked with the expression of anger and contempt, we assume that the facial expression of self-criticism includes the AUs defining anger and contempt. 

## 4. Facial Expression of Self-Protection

While self-criticism is defined as a form of problematic anger, protective anger (also known as assertive anger or adaptive anger) is acknowledged as a healthy anger [1,6,59]. According to Pascual-Leone and Paivio [59] and Pascual Leone [6], accessing adaptive anger by being self-protective is a healthy emotion and a form of self-defense that helps clients to assert themselves as long it maintains the adaptive purpose. Protective anger at maltreatment activates a sense of power that encourages clients to establish interpersonal boundaries [7,59]. However, under-regulation or overregulation of anger makes it difficult for clients to productively express their anger in a healthy way. In their work on EFT for anger in complex trauma, Pascual-Leone and Paivio (p. 40, [60]) define the following criteria for the effective expression of protective anger: Anger must be directed outward towards the perpetrator.Anger must be differentiated from other emotions (e.g., not being mixed with tears or fear).Assertive expression (e.g., using “I”-statements).The intensity of the anger expression must be appropriate to the situation.The anger expression must have a sort of explorative meaning.

In view of this, it is clear that the facial expression of self-protection discerns from the one of self-criticism. Although the effectiveness of self-protection is drawing more attention in psychotherapy research, a great majority of research so far has focused on the expression of rejecting unhealthy anger. As a result, to date, there are no studies examining the facial expression of self-protection. 

## 5. Aim of the Study

To the best of our knowledge to date, there are no studies examining clients’ in-session facial expressions of self-criticism, self-compassion, and self-protection. Therefore, the goal of this study is to explore the following question: Which facial AUs recognized by the computer software Affectiva determine the clients in-session facial expressions of self-compassion, self-protection, and self-criticism? 

**Hypothesis** **1.***On the basis of previous research and considering self-compassion as compassion directed inward, we hypothesize that the facial expression of self-compassion is one combined with the AUs of sadness (AUs 1 + 4 + 15) and happiness (AUs 6 + 12_smile)* [42].

**Hypothesis** **2.***In view of the fact that self-criticism is linked to the expression of anger* [10] *and contempt* [56]*, we assume that the facial expressions of clients being self-critical, recognized by Affectiva, is signaling facial AUs of anger (AUs 4, 5, 7, 24)* [36] *and contempt (AU12_smirk and AU43)* [58].

## 6. Final Research Question

Hitherto, there is no analysis on the facial expression of self-protection. For this reason, it is worthwhile to focus attention on the following question: Which facial AUs does Affectiva define for self-protection?

## 7. Methods

### 7.1. Material and Participants 

Owing to the consequence of ethical limitations and requirements needed for the analysis, we decided to examine videos of real EFT therapy sessions which can be purchased and are available for research. The video tapes needed to meet the following criteria: Must be an EFT therapy video.Involve sequences of clients expressing self-compassion, self-protection, or self-criticism during two-chair tasks within the therapy session.The quality of the tapes had to be sufficient for facial analysis.Uses the English language.

A total of seventeen EFT sessions were reviewed by the first author and a second author was consulted. Based on the previously mentioned requirements, twelve of these were identified as valid cases for further analysis. The therapy sessions were led by EFT-certified and well-known expert therapists. All clients were women who had the opportunity to have one or several therapy sessions with experienced therapists (Les Greenberg, Rhonda Goldman, and Ladislav Timulak). The concerns addressed by the clients in the sessions were real-life problems. 

### 7.2. The EFT-Videos Address the Following Topics

EFT over time. Psychotherapy in six sessions [60]. Sessions 3, 4 and 6 were selected for this study.EFT for depression [61].This is a series of two sessions with Dione who is suffering from depression. Both sessions were chosen for the study.Working with core emotion [62].Working with current and historical trauma [63].Working with social anxiety with Dawn [64]Narrative processes in EFT with Hannah [65].Case formulation in EFT. Addressing unfinished business [66].Transforming Emotional Pain: An Illustration of Emotion-Focused Therapy video with Claire [67].

### 7.3. Procedure

Out of the 12 selected videos, sections in which clients were expressed self-compassion, self-protection, or self-criticism within a two-chair or empty-chair dialogue were extracted for the analysis. Furthermore, the final video segments had to fulfil two preconditions: first, the faces had to be orientated to the camera and fill a great proportion of the frame in order for iMotions to pick up the head position and face location [68]; and second, the sequences had to be at least a few seconds long as the algorithm is trained on moving stimuli (identified in consultation with an iMotions consultant). Given the fact that the videos were not specifically designed for analysing facial expression, not all the video sections were of the required quality. As a result, the camera setting in some cases could not be adjusted to obtain a suitable face recording. In view of this, and considering the requirements, four of the reviewed EFT videos were appropriate for the facial analysis of self-criticism, six for self-compassion, and six for self-protection. The four segments for self-criticism were extracted from the following videos: session 6 of the EFT over time series, Narrative processes in EFT, session 1 of the EFT for depression series, and Case formulation in EFT. The six sequences for self-compassion were chosen from the following videos: session 3, 4, and 6 of the EFT over time series, Narrative processes in EFT, and session 1 and 2 of EFT for the Depression series. With regard to the six cases for self-protection, the sequences were selected from the following videos: Working with core emotion, Working with current and historical trauma, session 4 of the EFT over time series, Case formulation in EFT, and Narrative processes in EFT. For self-protection and self-criticism, the sections include all different clients. For self-compassion, the video segments consist of three different clients. The facial expression of one client has been analysed three times in three different sessions and another client two times in two different sessions. An emotion-neutral facial expression was extracted as a baseline either before (nine videos) or after (two videos) the two-chair intervention, depending on the camera view (clients’ faces needed to be upfront in order to be recognized by the software). As stated in previous research [19,69], clients’ level of emotional processing is at its lowest at the beginning of the session before the intervention. In addition, according to Rochman et al. [70], clients are more familiar with the setting at the end of the session after the intervention and therefore feel less anxious. The relevant sequences were cropped with the video editing program Wondershare (Version 10.2.3, Wondershare Technology, Shenzhen, China) [71] and converted into MP4 files for analysis with Any Video Converter [72]. The final video sections were then used for the facial AU extraction with iMotions. Eventually, the length of the segments for self-criticism was between 0:34 s and 01:25 min, for self-compassion between 0:32 s and 01:02 min, and for self-protection between 0:11 s and 0:58 s. In total, we had 02:30 min of self-criticism, 03:34 min of self-compassion, 03:02 min of self-protection, and 02:22 min of baseline. 

### 7.4. Measurement Instrument

We decided to measure the facial expressions of the selected cases automatically with the computer vision software iMotions, which is based on the FACS [35]. Whereas FACS has been a useful and reliable method for decades, coding by hand has proven to be demanding and especially difficult when analyzing the facial expressions of blended emotions. Furthermore, this coding requires certified coders who are trained intensively, and the coding itself is very much time consuming. Over the past decade, there has been an increasing number of research documenting the advancements of computerized facial recognition [35,73,74,75,76]. New software allows for fully automatic real-time facial expression identification [35,73,74] and captures the richness and complexity of facial expressions. To date, there are three computer software tools available that recognise AUs automatically: Facereader [74], iMotions Facet, and iMotions Affectiva (AFFDX) [35]. 

For the purpose of this study, we decided to use the biometric research platform iMotions [77]. The iMotions Affectiva software (called Affectiva in this study) is specifically designed for analysing videos. Affectiva algorithms identify basic emotions of the face by analysing changes of essential face characteristics. The validity of the software has been confirmed by Stöckli et al. [35]. In another study, Affectiva’s accuracy was compared to the effectiveness of facial electromyography (fEMG) [78]. Furthermore, there is a number of research using Affectiva for facial expression analysis [52,79,80]. However, to date, there are no studies using the Affectiva software for the facial analysis of clients in therapy videos. The aim of the study is to investigate which facial AUs the Affectiva software detects for self-compassion, self-protection, and self-criticism from the selected video sequences. 

After eliminating and collecting the video sequences of clients’ expression of self-compassion, self-criticism, and self-protection, the videos will be imported into the Affectiva software. The software recognizes the face, detects the features of the face, and finally classifies the features into 20 AUs [81]. For this study, we will examine all 20 AUs (Table 1). 

## 8. Data Analysis

The statistical analysis was conducted via the program R, version 4.0.5 [82] with installed package “lme4” [83]. Observations were grouped within individuals and AUs (repeated-measures design), so a multilevel model (25 respondents, 20 AUs, 4 states—baseline, self-criticism, self-compassion, self-protection) was fitted. We did not have any information about the age of each respondent and all of them were women, so these “fixed effects” were irrelevant in our case. To be sure our model had the best fit with our data, we did likelihood-ratio tests with Akaike’s Information Criterion (AIC) [84] and Bayesian Information Criterion (BIC) [85]. The first three models compared fit of the random effects (respondents and action units). The best fit had the model with both random effects in it because of the AIC and BIC comparison (lower values = better fit) with a significance of <2.2 × 10^−16^. After that, we compared the model with fixed effect “states”, which was comprised of the four research settings—(1) when the respondents were self-criticizing, (2) when they were using self-compassion, (3) when they were using self-protection, and (4) the baseline moment, when they were not involved in the two-chair technique (in other words, the neutral-emotion moment). Again, the better fit with our data had the more complex model with both random effects and the fixed effect with the significance of 2.2 × 10^−16^. We then plotted diagnostic tests to determine the non-normality of residuals, which proved that our model matched the data. The model, therefore, had three parameters: RT (variability among respondents), AU (variability among AUs—action units), treated as random effects, and parameter states (baseline, self-criticism, self-compassion and self-protection), treated as fixed effect. The response was binomial (absence/presence of an AU). For this reason, we used a logistic multilevel regression model. Absolute thresholds were set to 50 [81] to create binary data (<50 = 0, >50 = 1). We reported a conditional R^2^ measure (overall effect size) and variance of random effects. 

## 9. Results

The RT variance of the multilevel model was 1.88 and the AU variance was 10.61, so we assume that variance of AUs is a lot higher than variance RT (variance of respondents), meaning that differences among the AUs are far larger than individual differences among respondents. Conditional R^2^ is 0.76 (very high effect size). For self-criticism, AU1, AU2, AU7, and AU12_smirk are higher than the baseline whereas AU5, AU20, and AU43 are lower than the baseline. AU4, AU6, AU9, AU10, Au12_smile, AU14, AU15, AU17, AU18, AU24, AU25, AU26, and AU28 are not significant. For self-compassion, AU2, AU4, AU15, and AU18 are higher than the baseline whereas AU5, AU7, AU12_smile, AU24, AU25, and AU26 are lower than the baseline. AU1, AU6, AU9, AU10, AU12_smirk, AU14, AU17, AU20, AU28, and AU43 are not relevant. For self-protection, AU1, AU4, AU12_smirk, and AU18 are higher than the baseline whereas AU5, AU12_smile, AU14, AU17, AU20, AU24, AU25, and AU28 are lower than the baseline. AU2, AU6, AU7, AU9, AU10, AU15, AU26, and AU43 are not considerable. See Figure 1 for details on the frequency of the AUs (differences between states).

## 10. Discussion

The purpose of this study was to detect the facial AUs of self-criticism, self-compassion, and self-protection in EFT sessions recognized by the computer software iMotions. Our hypothesis that the facial expression of self-compassion is a combination of AUs of sadness (AUs 1 + 4 + 15 + 17) and happiness (AUs 6 + 12) is confirmed by our results. Among the AUs identified by iMotions, the AUs 4 (brow furrow), 15 (lip corner depressor), and 17 (chin raise) are in line with the ones acknowledged for sadness [36], whereas the AU12_smile (lip corner puller) is significant for happiness [36]. Moreover, our results are in agreement with previous research that identified, among others, the action units AU7 (lid tighten) and AU12_smile (bilateral lip corner puller) as the most recurrent in participants watching a compassionate moment [42]. The results were confirmed by a further study [38] acknowledging AU7 as one of the most frequent action units while watching compassionate stimuli. According to Goetz et al. [45], compassion is a blended emotion directed towards another’s distress, incorporating aspects of sadness and happiness that arouses a motivation to alleviate the suffering. In agreement with this, Eisenberg et al. [46] describe a sadness towards someone else’s negative state as empathic sadness that activates the need to help. One of the main goals of the two-chair work is helping clients to develop self-compassion towards their own pain caused by their critical voice [12,13]. This kind of sadness can be blended with anger at one’s self for being vulnerable [23]. Therefore, it is reasonable that our results, in line with Barankova et al. [42] and Kanovsky et al. [38], identified AU7 as part of the facial expression of compassion. Furthermore, compassion involves aspects of happiness [45]. In addition, along the lines with Baránková et al. [42] and Kanovský et al. [38], our results indicate AU12_smile (bilateral lip corner puller) is relevant for the expression of compassion that communicates a smile and is related to the expression of happiness [25,30]. In agreement with Ekman and Cordaro [86] and Haidt and Keltner [43], our findings highlighted the difficulty of distinguishing between the facial expressions of compassion and sadness. Hence, in our study, clients’ facial expressions of self-compassion display AUs significance for sadness (AUs 4 and 15) blended with Aus of anger (Aus 5, 7 and 24) and happiness (AU12_smile). Furthermore, the action units Aus 2 (brow raise) and AU 26 (jaw drop) are signals for the expression of surprise [23,36]. In line with Barankova et al. [42], we assume that clients are surprised when they unexpectedly start being compassionate to their own suffering after being critical to themselves. Hence, AU25 (mouth open) can also be connected to the experience of surprise as the mouth opens while the jaw drops. AUs 25 and 26 can also be signs of clients having to speak, which requires them to open their mouths. In addition, some people associate AU18 (lip pucker, also known as lips pursed) with the expression of disagreement [87]. In accordance with this, one of the main goals of the two-chair work is helping clients to develop an understanding for their own pain and strength to disagree with the criticism of their critical voice [13]. 

With regard to self-criticism, the results also support our hypothesis. Affectiva identified AUs 5 (eyes widen) and 7 (lid tighten) as the ones distinctive for the expression of anger [36] and the action units AU12_smirk (unilateral lip corner puller) and AU43 (eye closure), which are characteristic for contempt [36,58]. This is in agreement with previous research [36,58] recognizing the unilateral lip corner puller (AU12_smirk) typical for the expression of contempt. In addition, Halamová et al. [52] analysed the facial expressions of participants criticizing themselves in a two-chair technique. Their results showed that participants displayed, among others, action units AU14 (Dimpler) and AU43, which are significant features of contempt [58]. Confirming this, Whelton and Greenberg [56] asked highly self-critical and noncritical participants to criticize themselves in a two-chair setting after remembering an episode of failure. They observed that high self-critics showed more contempt than the control group. Clearly, research shows that the facial expression of self-criticism includes signals of contempt. Furthermore, our results verify that the facial expression of self-criticism includes features of anger. Ample research agrees on AU5 (eyes widen) and AU7 (lid tighten) as significant aspects of the expression of anger [23]. At the same time, the AUs 1, (inner brow raise), 2 (outer brow raise), 5 (eyes widen), 7 (lid tighten), and 20 (lip stretch) are closely related to fear [23,36]. Along these lines, in EFT theory, social anxiety is seen as a secondary emotion covering more vulnerable emotions such as shame and sense of worthlessness [88]. As a result, self-criticism is a relevant part of anxiety [89]. Therefore, our research seems to confirm that clients show fear while being critical of themselves. Taking on this implication, our study demonstrates that clients’ facial expressions of their inner critical voice signals AUs characteristic of the expressions of anger, contempt, and fear. 

Concerning our third research question, to the best of our knowledge, there is no research examining the facial expressions of self-protection. The combination of AUs analysed by iMotions displays that self-protection is a blended emotion combing AUs signaling sadness (AUs 1, 4, 17), happiness (AU12_smile), anger (AUs 4, 5, 24), contempt (AU12_smirk and 14), and fear (AUs 1, 4, 5, and 20). Our results are in line with the understanding of self-protection in EFT theory. In EFT, self-protection is acknowledged as a healthy, primary, adaptive anger towards maltreatment [6,7]. It is characterized through firmness, assertiveness, and empowerment. Therefore, it is predictable that the facial expressions of clients communicating self-protection involve aspects of anger and contempt towards their inner critical voice. Along with this, a sudden AU28 (lip suck) is evident for the beginning of anger as if the person has something hostile to say and is holding back [23]. In the two-chair activity, critical dialogue clients are encouraged to stand up for their needs [7,13]. There is a general agreement that when people start to feel stronger and assert themselves, it increases their happiness [90]. Presumably, this explains why the software identified AU12_smile, which is a significant feature of happiness. As well as self-compassion, we can assume that AU25 indicates clients talking, which requires an open mouth. In addition, similar to self-compassion, AU18 is a gesture of disagreement [87] of the more self-resilient and empowered clients towards their self-critic. The AU combination 1, 4, 5, and 20 is characteristic for the expression of fear [36]. In line with this, it is well-known that self-criticism activates a fear of failure [54,91]. Hence, we presume that in the two-chair dialogue clients experience fear after hearing the demands of their critical voice before they assert themselves through the expression of their unmet needs. Furthermore, the AU17 (chin raise) is typical for the expression of pride [92]. According to the EFT case conceptualization, the intention of the two-chair work is the development of a more confident, stronger self [13,93]. Thus, the pride expression has an adaptive function which signals a more positive sense of self. 

Finally, our results demonstrate the complex combination of AUs for each state. However, the findings show that it is relevant to discriminate between self-protection and self-compassion as primary, adaptive emotions. Within the two-chair dialogue, each state needs to be expressed and is significant to combat self-criticism. 

## 11. Limitations and Future Research

The main limitations of our research concern our video material. Predominately, it should be emphasized that our material was not of a high recorded quality and the faces were not directed towards the camera. Hence, clients’ faces were not centered enough for iMotions to detect all features accurately. Further research could ensure that the cameras are positioned in a way where clients’ faces are in front of the camera so that the software can capture all features and the recordings have better quality. Additionally, the segments for self-compassion, self-protection, and self-criticism were cropped from different timings in the therapy sessions. Within EFT research and practice, it is acknowledged that the sequence of expressing emotions has an impact on the level of emotional arousal [94] and therefore on clients’ facial expression. Considering the fact that we removed the analysed sequences throughout the sessions, we did not examine how the temporal order might have had an impact on the facial expressions. Future studies could examine how the timing of expressing these states in the therapy session might influence the AUs. Alternatively, researchers can investigate the AUs of clients’ self-compassion, self-criticism, and self-protection communicated in the same order. Moreover, it is important to mention that the number of clients and the length of the videos for each state varied. Due of the natural setting of the therapy sessions, we could not influence the length of time the clients expressed each state. Furthermore, for self-compassion, one client was analysed twice and one three times as they appeared in multiple therapy sessions Consequently, our material was not coherent. Researchers in this field could study the action units of these states with more consistent material. Though our goal was to explore the AUs in real therapy sessions, our material was previously studio recorded and therefore the conditions were not entirely natural. However, it is worth highlighting that these things are difficult to accomplish in real therapy sessions. With regards to iMotions, it should be mentioned that the software recognizes 20 out of 46 existing AUs. Consequently, our results are limited to the AUs detected by iMotions. It would be interesting to see, in further studies, which AUs are identified for each state if all 46 AUs were analysed. Finally, it should be taken into account that the reviewed material was unwantedly biased towards women. Lastly, it is important to highlight the low number of video sequences as a result of the poor quality of the recordings. Thus, further research could investigate the AUs of these states in men and a higher number of clients.

## 12. Conclusions

Our study examined the AUs of self-criticism, self-compassion, and self-protection in EFT therapeutic sessions using the software iMotions. Our results revealed that the facial expression of self-compassion combines AUs similar to sadness, happiness, and anger. For self-criticism, the software detected AUs characteristic of contempt, anger, and fear. With regard to self-protection, clients’ facial expressions display AUs signaling sadness, happiness, anger, contempt, and fear. This is a first study exploring the facial AUs of clients in real EFT-therapy sessions. Our findings have important implications for psychotherapy research and practice by helping therapists to have a better understanding of how self-compassion, self-protection, and self-criticism are nonverbally communicated.

## Figures and Tables

**Figure 1 ijerph-20-01129-f001:**
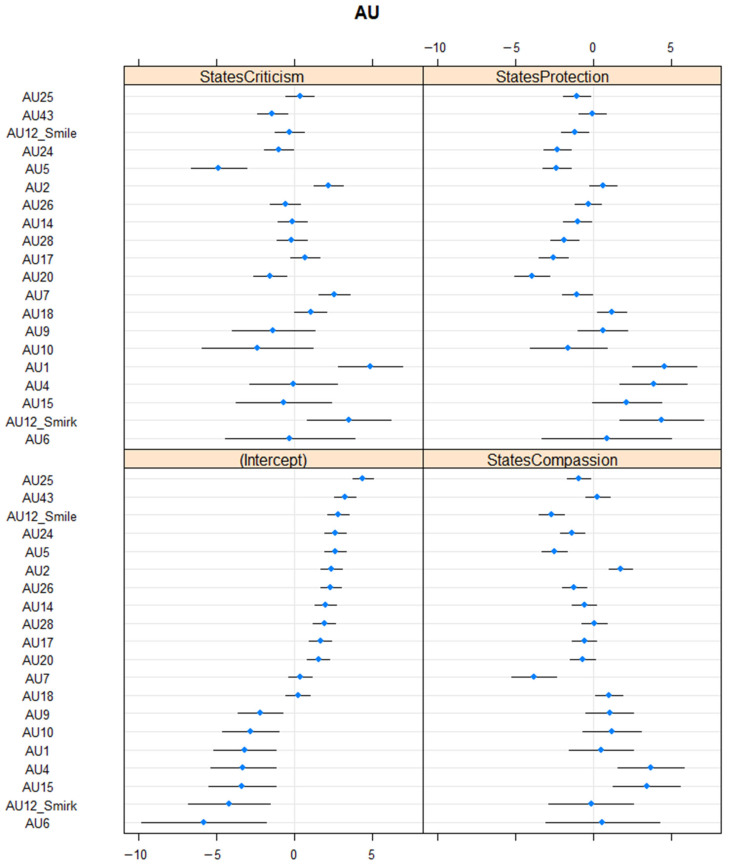
Frequency of Action Units—Differences Between States. Note. Intercept—baseline, StatesCompassion—self-compassion, StatesCriticism—self-criticism, StatesProtection—self-protection.

**Table 1 ijerph-20-01129-t001:** Action Units recognized by Affectiva.

AU Number	Affectiva
AU1	Inner Brow Raise
AU2	Outer Brow Raise
AU4	Brow Furrow
AU5	Eyes Widen
AU6	Cheek Raise
AU7	Lid Tighten
AU9	Nose Wrinkle
AU10	Upper Lip Raise
AU12	Smile
AU12	Smirk
AU14	Dimpler
AU15	Lip Corner Depressor
AU17	Chin Raise
AU18	Lip Pucker
AU20	Lip Stretch
AU24	Lip Press
AU25	Mouth Open
AU26	Jaw Drop
AU28	Lip Suck
AU43	Eye Closure

## Data Availability

In order to comply with the ethics approvals of the study protocols, data cannot be made accessible through a public repository. However, data are available upon request for researchers who consent to adhering to the ethical regulations for confidential data.

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
