# Peer review of "Clients’ Facial Expressions of Self-Compassion, Self-Criticism, and Self-Protection in Emotion-Focused Therapy Videos"

_ijerph, 2023, doi:10.3390/ijerph20021129_

Round 1

Reviewer 1 Report (Previous Reviewer 1)

Dear authors, thank you very much for your replies. I feel that the manuscript has improved. I have no further comments. Congratulations on the work and happy holidays.

This manuscript is a resubmission of an earlier submission. The following is a list of the peer review reports and author responses from that submission.

Round 1

Reviewer 1 Report

Dear authors, I have reviewed your manuscript entitled "Clients' Facial Expression of Self-Compassion, Self-Criticism, and Self-Protection in Emotion-Focused Therapy Videos". While they apply an interesting analysis, I believe that the proposal needs to be reviewed in depth. Here are some of the most important limitations I see:

- The introduction does not adequately discriminate which components specifically are to be measured within the therapy. On the one hand, the concepts of self-criticism, self-compassion and self-protection are defined as self-evaluations, i.e. as cognitive or thought-related attributes. However, elsewhere in the manuscript these concepts are defined as emotions. In this sense, I believe that there is no precise definition of the concepts of interest and it is not clear what the cognitive and what the affective-emotional components that are linked to these concepts consist of. Furthermore, it would be necessary to delimit how these components are supposed to be related to each other. For example, thoughts with self-critical content are associated with negative emotions, such as sadness. All of the above also calls into question the aim of the present proposal. As I see it, it is not the facial response to self-criticism, for example, that is being analysed, but rather the facial response linked to the emotion that generates this type of critical thinking towards oneself. All in all, the introduction, the objectives, and the contribution of the present proposal should be thoroughly revised.  

- Another aspect that I believe needs to be substantially improved is the method:

a- Authors should provide detailed and accurate information so that independent researchers can replicate the procedures applied. As I see it, it is not possible to replicate the present study with the information provided so far.

b- There are aspects of the method that need to be better explained and justified: - Regarding the selection of the videos, why those videos and not others; how was it determined that these were specific processes, for example linked to self-criticism; how does one know if they are "real" sessions or if they are mainly role playing? - Regarding the sample selection: it is not clear to me how many people end up being part of the analysis. For example, it is stated that four videos were used to analyse self-criticism, but it is not clear whether these were four sessions by the same person or four different people. In addition to this, I consider that working with such a small sample selected on the basis of ease of access severely limits the possibilities of generalisation and increases the likelihood of overfitting or models that are not easily extrapolated to other people. Related to this, the associated video-therapies show a great heterogeneity that does not seem to be contemplated later in the analyses (nor is it possible, I believe, given the small sample size): therapist; disorder treated; technique used; patient - Regarding the analysis: it is not clear to me how the baseline measurement was specified in each case, what determined that it was before or after the session? Furthermore, this variability should be contemplated in the linear model applied. More precision and justification should be given regarding the model applied and the specification of the variables as fixed or random effects. It would be important for the authors to provide the syntax of the procedure followed and specified in R. In addition, it would be appropriate to show initial exploratory analyses of the data.

I hope my comments are clear and will help to improve the report presented here. I wish you all the best in your work.

Author Response

Thank you for personally reviewing our manuscript and for giving us the opportunity to respond to your comments. We have made the relevant changes as detailed below and would like to resubmit our amended manuscript We hope you find our response detailed and satisfactory. If you have any further questions or comments, please do not hesitate to contact me.

Authors

Reviews 1

Review Report Form

Open Review

English language and style

( ) English very difficult to understand/incomprehensible
( ) Extensive editing of English language and style required
( ) Moderate English changes required
( ) English language and style are fine/minor spell check required
(x) I don't feel qualified to judge about the English language and style

Yes

Can be improved

Must be improved

Not applicable

Does the introduction provide sufficient background and include all relevant references?

( )

( )

(x)

( )

Are all the cited references relevant to the research?

( )

( )

(x)

( )

Is the research design appropriate?

( )

( )

(x)

( )

Are the methods adequately described?

( )

( )

(x)

( )

Are the results clearly presented?

( )

( )

(x)

( )

Are the conclusions supported by the results?

( )

( )

(x)

( )

Comments and Suggestions for Authors

Dear authors, I have reviewed your manuscript entitled "Clients' Facial Expression of Self-Compassion, Self-Criticism, and Self-Protection in Emotion-Focused Therapy Videos". While they apply an interesting analysis, I believe that the proposal needs to be reviewed in depth. Here are some of the most important limitations I see:

- The introduction does not adequately discriminate which components specifically are to be measured within the therapy. On the one hand, the concepts of self-criticism, self-compassion and self-protection are defined as self-evaluations, i.e. as cognitive or thought-related attributes. However, elsewhere in the manuscript these concepts are defined as emotions. In this sense, I believe that there is no precise definition of the concepts of interest and it is not clear what the cognitive and what the affective-emotional components that are linked to these concepts consist of. Furthermore, it would be necessary to delimit how these components are supposed to be related to each other. For example, thoughts with self-critical content are associated with negative emotions, such as sadness. All of the above also calls into question the aim of the present proposal. As I see it, it is not the facial response to self-criticism, for example, that is being analysed, but rather the facial response linked to the emotion that generates this type of critical thinking towards oneself. All in all, the introduction, the objectives, and the contribution of the present proposal should be thoroughly revised.  

Thank you very much for your comment. In this study we define according to EFT theory self-criticism as a maldaptive secondary anger and self-compassion and self-protection as primary adaptive emotions. I understand that the wording self-evaluation caused confusion. I made changes to make sure that it is clear that we talk about emotions in this research. Hence, the purpose of this study was to analyse the facial expression of clients while expressing self-criticism, self-compassion, and self-protection. In our understanding it is the facial response to self-criticism since we examined the facial action units while clients were being critical to themselves. Same for self-compassion and self-protection. We explained the sections „Facial expressions of self-compassion, self-protection, and self-criticism which emotions might be evoked in clients while expressing self-compassion, self-protection, or self-criticism.

- Another aspect that I believe needs to be substantially improved is the method:

a- Authors should provide detailed and accurate information so that independent researchers can replicate the procedures applied. As I see it, it is not possible to replicate the present study with the information provided so far.

b- There are aspects of the method that need to be better explained and justified: - Regarding the selection of the videos, why those videos and not others; how was it determined that these were specific processes, for example linked to self-criticism; how does one know if they are "real" sessions or if they are mainly role playing? - Regarding the sample selection: it is not clear to me how many people end up being part of the analysis. For example, it is stated that four videos were used to analyse self-criticism, but it is not clear whether these were four sessions by the same person or four different people. In addition to this, I consider that working with such a small sample selected on the basis of ease of access severely limits the possibilities of generalisation and increases the likelihood of overfitting or models that are not easily extrapolated to other people. Related to this, the associated video-therapies show a great heterogeneity that does not seem to be contemplated later in the analyses (nor is it possible, I believe, given the small sample size): therapist; disorder treated; technique used; patient - Regarding the analysis: it is not clear to me how the baseline measurement was specified in each case, what determined that it was before or after the session?

We described the selection procedure more in detail and added to which videos the selected segments were linked to. The clients were real cases. They were not actresses.Furthermore, It is now clear if the segments involved the same or different people. It was already explained why we chose baseline in some cases before and others after the workings phase: An emotion-neutral facial expression was extracted as a baseline either before (nine videos) or after (two videos) the two-chair intervention depending on the camera view (clients’ faces needed to be upfront in orderv to be recognized by the software).“

Finally, we are fully aware of the low number of cases as a limitation of this study. This is the first study that explores the facial expression of self-criticism, self-protection, and self-compassion. More studies involcing gender, culture, therapy method variety are needed to have a deeper understanding of the action units connected to self-criticism, self-compassion, and self-protection.

Furthermore, this variability should be contemplated in the linear model applied. More precision and justification should be given regarding the model applied and the specification of the variables as fixed or random effects. It would be important for the authors to provide the syntax of the procedure followed and specified in R. In addition, it would be appropriate to show initial exploratory analyses of the data.

Thank you for your comment. The baseline was the emotion-neutral facial expression of each participant so even if it was extracted before or after the session (only one moment for each participant), it did not need to be specified in the data or in the data analysis. I wrote the more detailed information about the whole procedure we followed in the text.

I hope my comments are clear and will help to improve the report presented here. I wish you all the best in your work.

Submission Date

17 September 2022

Date of this review

31 Oct 2022 14:36:33

Reviewer 2 Report

Abstract: Specify what the abbreviations AUs stand for. I recommend structuring the abstract in introduction, method, results, conclusions.

Introduction: It is recommended that the authors specify which theory of emotional change they are basing their research on.

In the paragraph "Facial expression of self-compassion" in the fourth line a period is missing after the word "open". I recommend rewriting the end of the paragraph since the reading becomes somewhat complicated, from the consideration of compassion as an emotion. Among other issues, the authors reflect compassion as a possible mixture of sadness and happiness, later stressing the need to differentiate between sadness and compassion, but why is it not meaningful to look for the difference between compassion and happiness?

In the paragraph "Facial expression of self-criticism" why does "(Anonymous, n.d.) " appear in line 11? Line 12 talks about the iMotions software without giving an explanation of what it is, who made it, how it works, etc. If this information is provided later it should be indicated. This suggestion is in line with the previous statement made by the authors regarding the mode of analyzing clients' facial expressions (automatic) at the end of the introduction. It should also be stated what they mean by that, even if it is something that will be explained later in the paper, for example with something like "...this study focuses on the automatic analysis through a computer software...". Also, in "aim of the study" the Affectiva software is mentioned for the first time. Is it different from iMotions, is it part of it, are they both used? This question is not clear either. It is also not explained what this software is and how it works.

The authors should better explain criterion 2 of "materials and participants". If the way to detect such emotions is through FACS - AUs it should be indicated.

Revise the numbering, tabulation and source in the paragraph "The EFT-Videos address the following topics".

I recommend putting the conclusions before the limitations.